# Utilization of Cooked Cassava and Taro as Alternative Feed in Enhancing Pig Production in Ecuadorian Backyard System

**DOI:** 10.3390/ani13030356

**Published:** 2023-01-19

**Authors:** Alfredo Valverde Lucio, Ana Gonzalez-Martínez, Evangelina Rodero Serrano

**Affiliations:** 1Faculty of Natural Sciences and Agriculture, University of the South of Manabí (UNESUM), Jipijapa 130303, Ecuador; 2Department of Animal Production, Faculty of Veterinary Sciences, University of Cordoba, AGR-134, ceiA3., 14071 Córdoba, Spain

**Keywords:** growth, small farms, subproduct feeds, tropical areas, swine

## Abstract

**Simple Summary:**

Pork production in Ecuador is a significant contributor to the country’s economy and food security, producing 227,769 metric tons of meat and 1,969,922 pigs, including 1,019,570 fattening pigs. This industry provides employment for 80,000 people, with many operations being family-run and utilizing alternative feed ingredients made from agricultural by-products or surplus crops to reduce costs. This approach to farming not only helps to lower production costs and increase profitability, but also minimizes environmental impact by using local resources and reducing waste. In addition, research has demonstrated that incorporating alternative feedstuffs, such as cooked cassava and taro, as partial substitutes for corn can lead to improved growth and fattening in pigs, as well as increased protein assimilation at the ileum level.

**Abstract:**

Pork production in Ecuador is of significant economic and nutritional importance. Many of these operations are family- or backyard-based and utilize alternative feed ingredients to reduce production costs. The current study aimed to determine the chemical composition of cooked cassava and taro, and to evaluate their inclusion in the feed of backyard pigs during the growth and fattening phases. A total of 42 castrated pigs from two geographic locations in Ecuador were studied over a period of 100 days, during which their weight and measurements were recorded at three-week intervals. At the end of the experiment, ileum samples were collected from the slaughtered pigs in order to calculate the apparent digestibility of the feed. The crude protein levels of cassava and taro were found to be 3.2% and 2.1%, respectively. The combination of cooked cassava and taro was found to be a suitable replacement for corn, with the best results observed in the group receiving a diet incorporating 21% each of cassava and taro. Analysis of the ileal content also revealed that this group exhibited the highest nitrogen assimilation from the diet.

## 1. Introduction

Meat production in Ecuador, with a national output of 227,769 metric tons and a pig population of 1,969,922, of which 1,019,570 are used for fattening, is a significant contributor to employment and food security in the country, generating 80,000 jobs [1]. The predominant production system in Ecuador is backyard family production, comprising 96% of the industry. There are also a few commercial (3%) and industrial (1%) farms [2].

The backyard pig farming system in Ecuador relies heavily on concentrated feeds, often (45% of pig producers) supplemented with alternative feed ingredients derived from household kitchen leftovers and agricultural crop by-products [3,4].

Feed accounts for over 65% of total production costs in pork production in Ecuador [5], and the most commonly used ingredients in concentrate preparation are cereals and oilseed meals [6], with corn making up 50–70% of cereals [7]. However, agricultural production generates residues that are not properly managed, leading to negative environmental consequences [8]. Ecuador generates agricultural residues from products such as sugar cane, rice, corn, bananas, and tubers such as potatoes [9]. Additionally, taro (*Colocacia esculenta (L.) Schott*) generates a non-exportable rejection of 7.5% of a production of 700,000 boxes per year [10].

Traditional farming systems, such as the backyard system, in Ecuador have the advantage of reducing costs through the use of alternative products in animal feed [11]. These systems are also of great economic significance for rural producers [12]. The introduction of alternative feedstuffs for pig rearing brings benefits in several areas: (i) economic and productive, as their nutritional content can replace traditional feeds without negatively impacting animal production [13,14,15] and reducing production costs [16,17]; (ii) food security, by reducing competition with other staple foods for human consumption [18]; and (iii) environmental, by reducing the disposal of crop residues and promoting crop diversification, which helps to decrease the environmental impact of corn production [18].

In tropical regions of Ecuador, alternative feed ingredients used in pig farming include cassava (*Manihot esculenta*), plantain (*Musa paradisiaca*), pumpkin (*Cucurbita maxima*), tagua (*Phytelephas aecuatorialis*), and taro (*Colocacia esculenta* (*L*.) *Schott*) [19]. To optimize the use of these feeds, it is important to analyze their nutritional content, suitability for animal feeding, and the physiological and/or productive stages of the animal [7]. Backyard producers typically use corn as the primary feed for pigs, which is supplemented with plantain, pumpkin, cassava, tagua, rice powder, and taro according to availability [20].

Taro grown in Ecuador is primarily exported to the USA and Puerto Rico due to a lack of consumption culture in the country of origin [21]. However, when the product does not meet size and shape standards for human consumption, it is used as animal feed [19]. Taro can be grown throughout the year in Ecuador’s tropical climate, resulting in an annual production of 17 metric tons [22].

Cassava is grown by small farmers in Ecuador year-round, at elevations ranging from near sea level to 1620 m above sea level. Ecuador has approximately 22,000 hectares [23] dedicated to cassava cultivation. It is frequently used for animal feed production, particularly the tuber, although the stem and leaves are also consumed fresh [24].

According to the Food and Agriculture Organization (FAO) [25], cassava is the fourth most important feed commodity after rice, wheat, and corn, with a starch content ranging from 55–77% [26], predominantly composed of amylopectin and a smaller amount of amylose. Cassava has a higher caloric content than other tubers, including potatoes [27]. Its low retrogradation starch content facilitates digestion and utilization by animals [24].

Taro has high-quality starch [17] due to its branched amylopectin, which facilitates the absorption and entry of water into intermolecular spaces, increasing solubility [28]. These characteristics make it suitable for feeding monogastric animals, such as pigs, and because taro starch is finely textured, it can be fed to young pigs [29].

According to previous research [30], cassava and taro primarily provide simple carbohydrates that are easily digestible for monogastric animals. However, further investigation is needed to determine their impact on animal health and growth [31]. These alternatives contain anti-nutritional factors or toxins that can negatively affect health and productivity, and should therefore not be fed raw [32,33]. Both cassava and taro contain hydrocyanic acid (HCN) [34], oxalates, and phytates [17,28,29,35], which can reduce the assimilation of nutrients [19] by binding with minerals such as calcium, magnesium, and iron, resulting in the formation of mineral salts that inhibit gut digestion [28]. While these components have not reached toxic levels, they can still affect animal health [28]. However, various treatments such as high temperature exposure or ensiling can significantly reduce the percentage of antinutrients in the feed [35,36,37,38], resulting in a more suitable feed for consumption [28,35,39].

Ecuadorian backyard pig producers in southern Manabí who sell animals directly for slaughter usually supply cooked cassava and taro as feed, supplemented with balanced feed [20]. In order to support the production and marketing of backyard pigs, it is crucial to understand their growth performance and standardize the most suitable feed formulation. Previous research has investigated the use of cassava or taro as alternatives to traditional pig feed [29,40,41], but there is very limited information available on the use of simultaneous cassava and taro formulations in pig feed.

The main goal of this study was to evaluate the effectiveness of using cassava and taro as alternative feed ingredients in the growth and fattening phase of pigs in order to enhance production conditions in Ecuadorian backyard pig-producing communities.

## 2. Materials and Methods

### 2.1. Geographical Location

This experiment was conducted in two locations in Ecuador: the Quinindé canton within the province of Esmeraldas and the Río Chico Parish within the Portoviejo canton of the province of Manabí (see Figure 1). These locations, which are situated in the Ecuadorian tropics, have average annual temperatures of 23–26 °C (00°13′33″ N and 73° 26′00″ E) and 25 °C (1°0′0″ S and 80°26′0″ W), respectively [42,43].

### 2.2. Data Collection and Experimental Diets

A total of 42 castrated crossbreed pigs (male and female) aged sixty days were used in the study. Before the experiment began, the pigs were given a 10-day period of adaptation to the location and a progressive change in their feed. It is worth noting that the volume of experimental feed was gradually increased every 5 days.

The pigs used in the study were creole pigs mixed with the Pretrain breed from small pig producers in the vicinity of the experimental location. The pigs selected for the study had similar weight at 45 days of birth, ensuring a high level of uniformity. The animals were transferred from local producers to the experimental location, where they were individually identified and vaccinated against African swine fever. They were given an acclimation period until they reached 60 days of age.

The animals were randomly divided into six groups (seven animals per group), corresponding to three different feeding diets (T1, T2, and T3) and two geographical locations (Quinindé, Ecuador and Río Chico, Venezuela). The pigs were housed in handmade pigsties with cement flooring, cement-plastered brick walls, zinc roofing (Figure 2), and a stocking density of 1.25 m^2^ per animal for the duration of the 100-day experiment. The pigsties had a trough large enough for all seven pigs in each group to feed simultaneously and a drinker with running water. The animals were exposed to natural light during the day and artificial light at night to protect against the common vampire bat (Desmodus rotundus). The temperature in both locations was similar throughout the trial.

The feed diets were formulated using the Excel program in the Microsoft Office 365^®^ suite, utilizing the scoring method [44]. The ingredients used were corn, rice powder, red palm oil, cassava, taro, and a protein concentrate for growth and fattening. Both cassava and taro were purchased fresh, with the tuber specifically used in the pig feed. The protein concentrate was sourced from the company “Bio alimentar” (Pelileo, Ecuador). This composition ensured that the pigs’ protein requirements, as well as their vitamin and mineral needs, were met.

To design complete diets, three replicates of 200 g of cassava tuber and three replicates of 200 g of taro tuber were analyzed in the chemical laboratory of the University of the South of Manabí (UNESUM) and the laboratory Multianalityca S.A. (Quito, Ecuador) (certified SAE LEN 09-008). This was done after the tubers were cooked for 30 min to determine their nutritional composition. After the chemical analysis of both feed alternatives was completed, the diets were formulated based on the nutritional needs of the pigs according to their reproductive stage (i.e., growth and fattening). The feed without corn replacement was provided in meal form, while those with cassava and taro were given wet. The cassava and taro were chopped, cooked for approximately thirty minutes or until softened, and salted. The diet with alternative feed ingredients was prepared daily, divided into two portions, and provided twice a day at 8 am and 3 pm for the entire 100-day duration of the experiment.

At the start of the experiment, the pigs were given 8.96 kg of feed per week, which was increased weekly according to the Genetiporc table for fattening pigs [45]. This resulted in a total of 22.47 kg of feed being provided in the final week. It was not necessary to take into account the feed that was not consumed, as all batches were fully consumed by the pigs. There were no instances of mortality among the 42 animals during the trial, and all remained healthy throughout the experiment.

### 2.3. Measurement for Production Performance and Digestibility

The productive performance of the animals was measured through live weight (kg), height (cm), and length (cm) of the animals, as well as their feed conversion ratio. Weight, height, and length were obtained individually by placing the animals in a cage that had previously been placed on a high-precision digital scale (Montero TCS300JC61Z (Quito, Ecuador), with a maximum capacity of 300 kg and a minimum of 2000 g (d = 100 g)). The feed conversion was calculated in each group of animals (T1, T2, and T3) using the ratio between the feed administered and the average weight gain [45]. The weighing and measurement (height and length) of the animals was carried out every 21 days, for a total of five data collections, three during the rearing phase and two during the fattening phase.

At the end of the experiment, the pigs were slaughtered in accordance with Ecuadorian animal welfare regulations [46], which involve stunning and bleeding after 10 h of fasting. To analyze the apparent digestion, a sample of 150 g of ileum was collected from 30 pigs (5 animals per group) and subsequently frozen at −20 °C for a period of less than 5 days [31]. The following characteristics were analyzed in each ileum sample (the method of analysis is specified in brackets): moisture (Association of Official Agricultural Chemists (AOAC) 925.10); crude protein (AOAC 2001.11); fat (AOAC 2003.06); ash (AOAC 923.03); pH (Ecuadorian Technical Standard (NTE) INEN ISO 4316:2014m); crude fiber (NTE INEN 522:2013). The calories, carbohydrates, and dry matter of the ileum were estimated through the following calculations, based on methods established by Maclean et al. [47]:

Calories = (carbohydrate (g) × 4) + (protein (g) × 4) + (fat (g) × 8) + (fiber (g) × 4);

Carbohydrate = 100 − (moisture + fiber + fat + protein + ash);

Dry matter = (initial weight − dry weight)/initial weight.

The determination of the apparent digestibility was carried out by applying the following equation of Lachmann and Febres [48], as adapted by Pico Dominguez [49]:$$\mathrm{Coefficient}\;\mathrm{of}\;\mathrm{digestibility}\;(\%)=\frac{\left(NI-NH\right)}{NI\times 100}$$

*NI* (nutrient ingested) represents 15% of the protein content of the diet received by the animal at the end of the trial, while *NH* (nutrient in ileum) is the percentage of protein from the chemical analysis of the ileum.

### 2.4. Economic Analysis

The economic benefits of using alternative feeds in pig rearing and fattening were evaluated. This process took into account the initial cost of acquiring and transporting the animals, as well as the cost of setting up housing facilities for the animals. Feeding and health costs during the experiment, as well as costs related to slaughtering and cleaning utensils, were also considered. The economic analysis was conducted using a cash-flow approach, in which costs and revenues were estimated, resulting in the application of the financial ratio benefit–cost ratio and the unit cost of pork according to the treatments [50].

### 2.5. Statistical Analysis

The data were analyzed using two completely randomized Analysis of Variance (ANOVA) models: one for body traits and feed conversion and one for ileal traits. For the analysis of the effects of body measurements and feed conversion, a factorial arrangement of balanced repeated measures was applied. The model included the variables of geographic location (L), treatment (T), time factor (FT), and their interaction (L × T × FT) (Equation (1)). For the model with an ileum chemical analysis, the model included variables of geographical location (L), treatment (T), and their interaction (L × T). The LSD test was used to compare LSMEANS, with a significance level set at *p* < 0.05.
Y = µ + L_i_ + T_j_ + FT_k_ + (LT)_ij_ + (LFT)_ik_ + (TFT)_jk_ + (LTFT)_ijk_ + ε_ijk_(1)
where: µ is the overall mean, L is the geographical location, T is the feeding or diet system treatment, FT is time factor, LT is the interaction location and feeding, LFT is the interaction location and time factor, TFT is the interaction feeding and time factor, LTFT is the interaction location, feeding and time factor, ε is the sampling error, and ijk is for any value of ijk.

To further analyze the effects of the various factors on body traits and feed conversion, regression analyses were performed. The economic parameters were also analyzed.

Statistical analysis was conducted using Statistica 12.0 for Windows and InfoStat 2020e software.

## 3. Results and Discussion

### 3.1. Nutritional Composition of Cooked Cassava and Taro as Feed Alternatives to Replace Corn for Backyard Pigs

The chemical analysis of cooked cassava and taro is presented in Table 1. The analyses showed that cooked cassava had a crude protein content of 3.2%, while taro had a crude protein content of 2.1%. The crude fiber content was similar in both tubers (1.75 in cassava and 1.53 in taro), while the lipid content was higher in cassava (0.73 versus 0.18).

The protein levels we have obtained in cooked cassava (3.2) are lower than those obtained by Rodríguez et al. [51], as well as by Suárez and Mederos [24] (5.17) in uncooked cassava, but higher than those provided by Vargas and Hernandez [27]. These differences may be due to the fact that the protein levels of the cassava tuber depend on the variety and the type of management during cultivation. This is especially the case regarding that which affects fertilization [24]. In addition, the lipids obtained for cassava showed similar values to those obtained by Vargas and Hernández [27].

According to Knowles et al. [26] (whose results are consistent with ours in Table 1), the crude fiber content is low. Our sample of cooked taro had a crude fiber content of 1.53 g/100 g, similar to the results of Rodríguez-Miranda et al. [52] (1.56 g/100 g), but lower than that reported by Púa et al. [17]. The latter authors attributed differences in the results of different studies to the degree of maturity at which the tuber is harvested. However, our sample of taro had similar crude protein, lipid, and ash content to that reported by Púa et al. [17]. According to this author, magnesium is the predominant mineral in taro, followed by calcium, iron, and zinc. Other researchers have found similar lipid and ash content in taro samples, but higher protein and fiber content [28,29].

Based on the data from the chemical analysis, the composition of the diet administered to each of the treatments (T1, T2, and T3) was determined. This was done for both the growth and fattening phases of the pigs, as shown in Table 2. The diet administered to the pigs in the T1 or control group did not contain any cassava or taro, while the other two groups received 32% (T2) and 42% (T3) of cassava and taro in equal amounts, based on previous studies [40,53]. Both alternatives substituted a portion of the corn in the feed, so that the amount of corn was reduced by 60% in T2 and 81% in T3 by 81%.

### 3.2. Growth and Productive Performance of Backyard Pigs in Ecuador Reared on Cooked Cassava and Taro Feed Alternative to Corn

The pigs had an average starting weight of 13.66 ± 0.49 kg and showed exponential growth throughout the experiment, with the T3 group (42% alternative feed), exhibiting particularly strong growth compared to the other two groups (see Figure 3).

Although the two phases were distinguished based on the productive stage of the animals (growth to fattening), the regression curve of the weight and size of the animal (height and length) in relation to the feed conversion ratio did not show an inflection point corresponding to the change in stage.

Geographical location significantly influenced pig growth (*p* < 0.05) (Table 3). These differences may be attributed to the genetic diversity of the pigs, which results from the various combinations of breeds, leading to varied responses [54]. Pigs that received a diet containing 42% feed alternative (T3) from Quinindé exhibited the highest weight gain throughout the growth phase (12.35 kg between control 1 and 2; 19.14 kg between control 2 and 3) (Table 4). In contrast, the poorest growth data was observed in T1 (8.39 kg between control 1 and 2) and T2 (11.58 kg between control 2 and 3), both from Río Chico. Liveweight was statistically different (*p* < 0.01) among the controls in each of the six groups, as well as among the treatment and geographical location variables. Pigs in the T1 group from Quinindé were the largest at the beginning of the rearing phase (37.71 cm in control 1 and 44.29 cm in control 2), but size was improved in T3 from the same location at the end of the rearing phase (50.71 cm in control 3). In terms of body elongation, pigs in the T3 group exhibited the greatest increase in length over the course of the first phase (*p* < 0.01). Despite the positive growth performance of pigs fed cassava and taro alternatives, feed conversion ratio was better in the T1 (conventional) pig groups for both geographical locations, Quinindé and Río Chico at 1.4 and 1.8, respectively. Additionally, pigs in the T3 (42% feed alternative) group had the worst feed conversion ratios (2.2 and 2.0, respectively).

Geographical location significantly (*p* < 0.05) affected pig growth during the fattening phase (Table 5). These differences may be due to genetics, as the miscegenation of pig breeds can result in varied responses [54]. Pigs fed with 42% feed alternative (T3) from Quinindé showed the lowest weight gain during the fattening phase, with an increase of 23.68 kg, while pigs from Río Chico and those fed with conventional concentrate increased the most in weight, by 36.19 kg. In terms of height, pigs from Quinindé fed with a 42% feed alternative showed the smallest increase, at 3.00 cm, while pigs from Río Chico and those fed with conventional concentrate increased the most, by 8.28 cm. Similarly, pigs from Quinindé fed with a 42% feed alternative showed the smallest increase in length, at 5.43 cm, while pigs from Río Chico in the T2 group showed the largest increase, at 14.29 cm. In terms of feed conversion ratio, as in the previous stage, pigs from Quinindé fed with conventional feed (T1) showed the best results in the two control groups, with ratios of 2.13 and 3.05. The worst feed conversion ratio was observed in the T3 group from Quinindé at the beginning of the fattening phase (2.79) and in the T1 group from Río Chico at the end of the fattening phase (3.73).

Previous studies involving the substitution of raw cassava for corn as feed have demonstrated positive results. Romero et al. [40] observed that up to 30% replacement of corn with cassava bran resulted in a weight gain of 36 kg in the fattening stage, which was 0.5 kg higher than the control group, and a 3.4% improvement in feed conversion rate. It should be noted, however, that these values are higher than those observed in the present study.

Cooked taro has been found to be capable of substituting up to 50% of corn in pig feed [17], with some research indicating that the substitution can reach 25% when taro is fed raw [17]. Other research suggests that taro fed as silage can completely replace corn [41] without impacting the productive parameters of the pigs [19,29].

Our findings indicate that the most productive yields were obtained when corn was replaced with a combination of 42% cooked cassava and taro (21% of each tuber). Thus, the incorporation of alternative feedstuffs in both the growth and fattening phases of the pigs did not result in any losses, and the T3 formulation was an excellent choice in terms of productive outcomes.

### 3.3. Ileal Apparent Digestion in Backyard Pigs Fed Cassava and Taro to Replace Corn

The chemical analysis of ileum content showed no significant differences (*p* > 0.05) in the six groups of pigs, except for the percentage of ash, which was higher in the T3 group from Quinindé (3.31%) (Table 6).

The process of ileal apparent digestion is the difference between the ingested nitrogen and the nitrogen content of the ileum [48]. Intestinal contents are composed of a mixture of dietary and endogenous proteins, which are derived from digestive secretions and desquamated plant cells [55]. The results of a trial carried out to assess ileum digestion revealed that there were no significant differences (*p* > 0.05) in the percentage of protein content in the ileum between the treatments, with values ranging from 56.13% (T3) to 65.13% (T1). The pigs in the T2 and T1 groups recorded the highest percentages of protein content in the ileum, especially the Río Chico animals (Table 7). However, the pigs fed with higher amounts of cassava and taro (T3) showed higher nitrogen assimilation, with values of 56.13% and 57.20%, respectively.

There is no evidence of apparent ileal digestion of either cooked cassava or taro when fed simultaneously. However, the values obtained in our trials were higher than those reported by Almaguel et al. [30], who found an ileal digestion of 43.9% when pigs were fed with cassava foliage. This may be attributed to the higher fiber content of the forage. On the other hand, Parra et al. [53] reported an apparent rectal digestion of 74.58% for pigs fed with cassava meal obtained from the root and foliage. Regarding taro, the level of digestibility in the rectum of taro meal and silage found by Caicedo et al. [19,41] was 91.89 and 87.81%, respectively.

### 3.4. Economic Analysis of Treatments

An important consideration when selecting feed alternatives is the potential to reduce production costs. Table 8 presents the summary of the costs associated with rearing and fattening the pigs, as well as the profit obtained after their sale. The highest value for the benefit/cost ratio was found for the T3 group from Quinindé (1.77), as this group had the lowest production cost per kilogram of meat (USD 3.61). The worst economic indicators were obtained for the T2 pigs from the same location, with a benefit/cost ratio of 1.46 and a total production cost per kilogram of meat of USD 3.91. These findings suggest that the T3 group, with a cost of 3.61 USD/kg of meat, generates an average saving of 0.07 US cents for each kilogram of meat produced, thereby generating higher profits when compared to T1. These results are in agreement with the findings of González [6], who states that the use of feed alternatives such as cassava can reduce production costs by up to 23.5% without compromising productivity [40]. For example, the use of cassava bran reduced production costs by 5 US cents per kilogram of feed produced, which is a significant reduction in pig production. Additionally, Aragadvay et al. [56] reported greater profits when using a 30% taro diet.

Feed efficiency is a critical component of agricultural economics, as it offers the potential to increase producer income [57]. In the context of production in the Ecuadorian tropics, a common practice is to replace corn with cassava in the diet of pigs during the fattening growth stages [4,58]. Cassava can completely replace corn as a feedstock [6,34], although it is important to control the level of foliage offered and the fiber content, which can reach up to 25.97% [37], as this could negatively affect the animals, increasing the number of peristaltic movements and accelerating the rate at which the feed passes through the digestive system [16]. In this study, we investigate, for the first time, the possible use of a combination of cassava and taro in animal feed. Our results demonstrate that, when both are cooked and administered on the same day (as is customary among backyard pig farmers [20]), cassava and taro can effectively replace corn in diets of up to 45% without any losses (i.e., no deaths or disease). Furthermore, we observe a considerable improvement in yields, along with a decrease in costs. The utilization of local feed alternatives reduces environmental impact and supports the development of sustainable and integrated agriculture [15,59].

## 4. Conclusions

The combination of cooked cassava and taro, administered in equal proportions, and following the traditional practice of Ecuadorian backyard pig producers, has been shown to have a chemical composition suitable for use as an alternative in pig feed, allowing for a reduction of almost 20% in the amount of corn in the diet.

The combined use of cooked cassava and taro in the feed formulation for backyard pigs has been demonstrated to be a valid alternative to corn. In terms of production performance during growth and fattening, the assimilation of dietary nitrogen, reduced dependence on the rearing environment, and the inclusion of 42% cassava and taro in the formulation is an effective option.

The combined use of cooked cassava and taro drawn from crop surpluses or by-products in the feeding of backyard pigs in Ecuador contributes to the circular economy by significantly reducing production costs, thereby improving the average benefit/cost balance. Additionally, it also helps to reduce environmental impacts by utilizing local inputs and reducing waste production.

## Figures and Tables

**Figure 1 animals-13-00356-f001:**
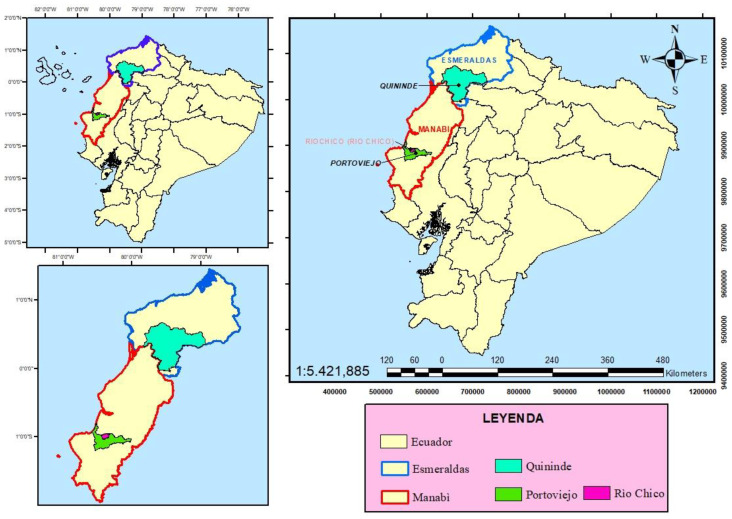
Geographic location of the sampling zones. Produced by: Ing. Juan García Cabrera. Mg. Sc and Ing. Alfredo Valverde Lucio. Mg. Sc.

**Figure 2 animals-13-00356-f002:**
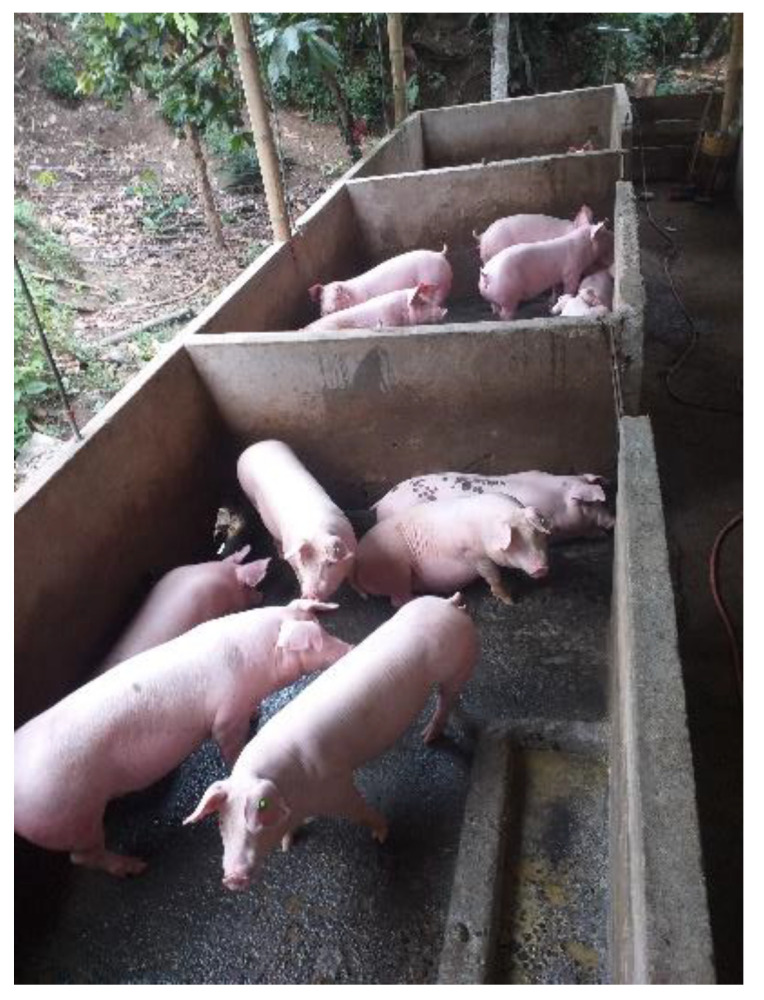
Pigs used and the pigsties employment in the trial.

**Figure 3 animals-13-00356-f003:**
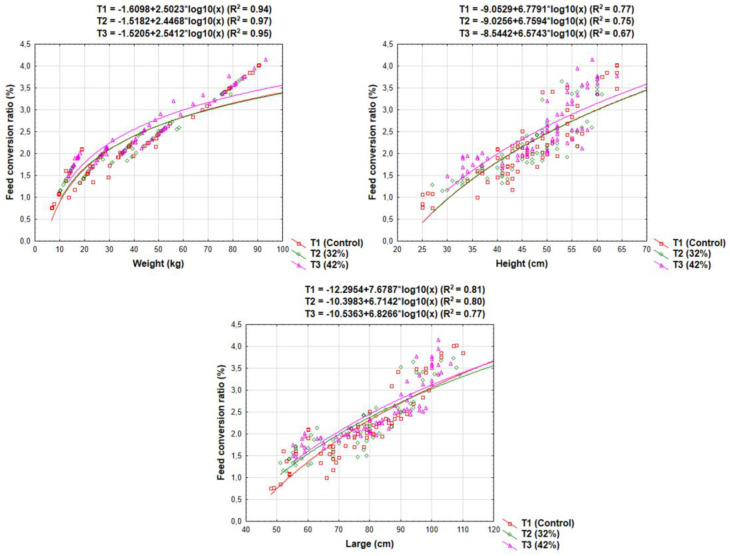
The relationship between body traits and feed conversion ratio in each treatment. T1: conventional feed with no corn replacement; T2: corn replacement with 32% of cassava + taro; and T3: corn replacement with 42% of cassava + taro.

**Table 1 animals-13-00356-t001:** Nutritional composition of cooked cassava and taro.

Parameters	Cooked Cassava	Cooked Taro
Dry matter (%)	37.86	39.15
Protein (%)	3.20	2.10
Humidity (%)	62.24	60.85
Ash (%)	1.04	1.11
Crude fiber (%)	1.75	1.53
Lipid (%)	0.73	0.18
Organic matter (%)	36.80	38.04

Ash is the powdery residue left after the burning of a substance.

**Table 2 animals-13-00356-t002:** Nutritional value and composition of the used diets.

Ingredients ^1^	Phase					
	Growth			Fattening		
	T1 (Control)	T2 (32%)	T3 (42%)	T1 (Control)	T2 (32%)	T3 (42%)
Corn (kg)	23.64	9.55	4.55	23.64	9.55	6.36
Protein concentrate (kg) ^2^	13.18	16.36	17.27	11.36	15.00	16.36
Rice powder (kg)	8.18	4.55	4.09	10.00	5.91	3.18
Cooked cassava (kg)		7.27	9.55		7.27	9.55
Cooked taro (kg)		7.27	9.55		7.27	9.55
Red palm oil (kg)	1	1	1	1	1	1
Salt (g)	0.25	0.25	0.25	0.25	0.25	0.25
Crude protein (%)	17.35	17.35	17.35	15.30	15.29	15.29
Gross energy (kcal/kg)	3098	3079	3077	3084	3079	3094

^1^ Quantity per kilo of feed. ^2^ Soybean meal, rice by-products, banana meal, free fatty acids, molasses, calcium carbonate, mycotoxin binders, vitamin supplements (A, D3, E, and K3), riboflavin, niacin, thiamine mononitrate, cyanocobalamin, pyridoxine hydrochloride, biotin, trace mineral supplements, manganese sulfate, zinc sulfate, copper sulfate, ferrous sulfate, sodium, calcium iodide, methionine, lysine (such as hydrochloride and sulfate), threonine, choline chloride, antifungals, enzymes, antibiotics, and antioxidants.

**Table 3 animals-13-00356-t003:** Repeated measures analysis for the productive performance parameters.

Source of Variation	DF	Traits			
		Weight	Height	Length	Feed Conversion Ratio
Treatments (T1, T2 and T3)	2	*p* < 0.001	*p* < 0.001	*p* < 0.001	*p* < 0.001
Location (Quinindé and Rio Chico)	1	*p* < 0.001	*p* < 0.001	0.608	*p* < 0.001
Time point (1 to 5)	4	*p* < 0.001	*p* < 0.001	*p* < 0.001	*p* < 0.001
Location × Treatments	2	*p* < 0.001	*p* < 0.001	*p* < 0.001	*p* < 0.001
Time point × Treatments	8	0.514	*p* < 0.01	0.062	0.516
Location × Treatments × Time point	8	0.784	0.678	*p* < 0.01	0.912
Error	161				
Total	204				

T1: conventional feed with no corn replacement; T2: corn replacement with 32% of cassava + taro; and T3: corn replacement with 42% of cassava + taro.

**Table 4 animals-13-00356-t004:** Productive performance (mean ± standard error (coefficient of variation)) of growing backyard pigs fed with different formulations from two locations in Ecuador (Quinindé and Río Chico).

Traits	Time Point	Quinindé			Río Chico			P		
		Treatments ^1^								
		T1 (Control)	T2 (32%)	T3 (42%)	T1 (Control)	T2 (32%)	T3 (42%)	Location (L)	Treatment (T)	L × T
Weight (kg)	1	8.43 ± 0.6 (3.81) ^C,c^	11.43 ± 0.3 (7.03) ^b,b^	15.96 ± 0.63 (10.5) ^C,a^	15.40 ± 1.07 (18.11) ^C,a^	14.39 ± 0.44 (4.67) ^C,a^	15.61 ± 0.64 (10.86) ^C,a^	*p* < 0.001	*p* < 0.001	*p* < 0.001
	2	18.2 ± 1.22 (16.47) ^B,d^	20.39 ± 0.58 (7.51) ^B,c,d^	28.31 ± 0.65 (6.07) ^B,a^	23.79 ± 0.79 (8.75) ^B,b,c^	25.22 ± 1.2 (12.64) ^B,a,b^	26.45 ± 0.92 (9.24) ^B,a,b^	*p* < 0.001	*p* < 0.001	*p* < 0.001
	3	33.13 ± 2.27 (16.79) ^A,b^	38.53 ± 1.58 (10.87) ^A,b^	47.45 ± 1.7 (9.5) ^A,a^	35.86 ± 0.83 (6.13) ^A,b^	36.80 ± 1.12 (8.02) ^A,b^	38.34 ± 0.95 (6.55) ^A,b^	0.028	*p* < 0.001	*p* < 0.001
	P	*p* < 0.001	*p* < 0.001	*p* < 0.001	*p* < 0.001	*p* < 0.001	*p* < 0.001			
Height (cm)	1	25.83 ± 0.4 (3.81) ^B,d^	30.71 ± 0.84 (7.21) ^C,c^	33.29 ± 0.68 (5.41) ^C,b,c^	37.71 ± 0.89 (6.26) ^C,a^	35.86 ± 0.63 (4.67) ^C,a,b^	35.57 ± 0.75 (5.59) ^C,a,b^	*p* < 0.001	*p* < 0.01	*p* < 0.001
	2	41.17 ± 1.08 (6.41) ^B,b^	44.43 ± 0.3 (1.9) ^B,b^	45.71 ± 0.42 (2.43) ^B,a^	44.29 ± 1.04 (6.21) ^B,a,b^	42.43 ± 1.23 (7.68) ^B,a,b^	44.29 ± 0.9 (5.33) ^B,a,b^	0.225	*p* < 0.01	0.051
	3	42.17 ± 0.87 (5.07) ^A,c^	46.43 ± 0.65 (3.7) ^A,b^	50.29 ± 0.47 (2.49) ^A,a^	49.29 ± 0.61 (3.25) ^A,a^	50.14 ± 0.67 (3.54) ^A,a^	50.71 ± 0.29 (1.49) ^A,a^	*p* < 0.001	*p* < 0.001	*p* < 0.001
	P	*p* < 0.001	*p* < 0.001	*p* < 0.001	*p* < 0.001	*p* < 0.001	*p* < 0.001			
Length (cm)	1	51.57 ± 1.12 (5.29) ^C,c^	54.31 ± 1.15 (5.56) ^C,b,c^	61.29 ± 1.64 (7.09) ^C,a^	56.71 ± 1.29 (6.00) ^C,a,b^	55.29 ± 0.36 (1.72) ^C,b,c^	56.86 ± 0.67 (3.12) ^C,a,b^	0.667	*p* < 0.001	*p* < 0.001
	2	66.17 ± 0.75 (2.77) ^B,b,c^	71.14 ± 2.51 (9.34) ^B,a,b^	77.57 ± 1.21 (4.13) ^B,a^	72.14 ± 1.68 (6.17) ^B,a,b^	62.29 ± 1.21 (5.14) ^B,c^	70.14 ± 1.45 (5.49 ^B,b^	*p* < 0.05	*p* < 0.001	*p* < 0.001
	3	75.83 ± 1.82 (5.86) ^A,b^	77.86 ± 0.99 (3.35) ^A,b^	90.71 ± 1.08 (3.16) ^A,a^	80.43 ± 1.36 (4.47) ^A,b^	76.14 ± 1.52 (5.28) ^A,b^	80.71 ± 0.99 (3.26) ^A,b^	*p* < 0.05	*p* < 0.001	*p* < 0.001
	P	*p* < 0.001	*p* < 0.001	*p* < 0.001	*p* < 0.001	*p* < 0.001	*p* < 0.001			
Feed conversion ratio (%)	1	0.94 ± 0.07 (17.55) ^C,c^	1.28 ± 0.03 (6.97) ^C,b^	1.78 ± 0.07 (10.44) ^C,a^	1.72 ± 0.12 (18.29) ^B,a^	1.6 ± 0.05 (8.09) ^C,a^	1.74 ± 0.07 (10.83) ^B,a^	*p* < 0.001	*p* < 0.001	*p* < 0.001
	2	1.34 ± 0.09 (16.43) ^B,d^	1.5 ± 0.04 (7.47) ^B,c,d^	2.08 ± 0.05 (6.14) ^B,a^	1.75 ± 0.06 (8.77) ^B,b,c^	1.86 ± 0.09 (12.76) ^B,a,b^	1.95 ± 0.07 (9.23) ^B,a,b^	*p* < 0.001	*p* < 0.001	*p* < 0.001
	3	1.9 ± 0.13 (16.70) ^A,b^	2.21 ± 0.09 (10.09) ^A,b^	2.72 ± 0.1 (9.48) ^A,a^	2.06 ± 0.05 (6.23) ^A,b^	2.11 ± 0.06 (8.01) ^A,b^	2.2 ± 0.5 (6.46) ^A,b^	*p* < 0.05	*p* < 0.001	*p* < 0.001
	P	*p* < 0.001	*p* < 0.001	*p* < 0.001	*p* < 0.001	*p* < 0.001	*p* < 0.001			

^1^ T1: conventional fed with no corn replacement; T2: corn replacement with 32% of cassava + taro; and T3: corn replacement with 42% of cassava + taro. ^A,B,C^ for each group within a geographical location, as well as the least square means without a common superscript differ significantly (*p* < 0.05) between time point. In addition, ^a,b,c,d^ for each control, least square means without a common superscript differ significantly (*p* < 0.05) between groups.

**Table 5 animals-13-00356-t005:** Productive performance (mean ± standard error (coefficient of variation)) of backyard pigs fed in fattening stage with different formulations of cassava and taro in two locations in Ecuador (Quinindé and Río Chico).

Traits	Time Point	Quinindé			Río Chico			P		
		Treatments ^1^						Location (L)	Treatment (T)	L × T
		T1 (Control)	T2 (32%)	T3 (42%)	T1 (Control)	T2 (32%)	T3 (42%)			
Weight (kg)	1	43.18 ± 3.12 (17.7) ^b^	47.60 ± 2.40 (13.36) ^a,b^	56.73 ± 2.58 (12.04) ^a^	47.72 ± 1.79 (9.91) ^a,b^	48.93 ± 2.11 (11.41) ^a,b^	48.99 ± 1.39 (7.50) ^a,b^	0.735	*p* < 0.01	*p* < 0.05
	2	68.62 ± 4.67 (16.66) ^b^	71.64 ± 3.74 (13.87) ^a,b^	80.41 ± 3.13 (10.31) ^a,b^	83.91 ± 2.87 (9.04) ^a^	78.93 ± 3.26 (4.22) ^a,b^	80.38 ± 3.19 (3.91) ^a,b^	*p* < 0.01	0.200	0.051
	P	*p* < 0.001	*p* < 0.001	*p* < 0.001	*p* < 0.001	*p* < 0.001	*p* < 0.001			
Height (cm)	1	44.17 ± 0.87 (4.84) ^d^	48.71 ± 0.42 (2.28) ^c^	52.86 ± 0.59 (2.98) ^b^	53.86 ± 1.01 (4.96) ^a,b^	56.00 ± 0.49 (2.31) ^a^	56.00 ± 0.53 (2.53) ^a^	*p* < 0.001	*p* < 0.001	*p* < 0.001
	2	51.83 ± 1.19 (5.65) ^c^	54.43 ± 1.27 (6.17) ^c^	55.86 ± 0.74 (3.49) ^b,c^	62.14 ± 1.12 (4.78) ^a^	60.14 ± 0.14 (0.63) ^a^	59.86 ± 0.83 (3.66) ^a,b^	*p* < 0.001	0.654	*p* < 0.01
	P	*p* < 0.001	*p* < 0.001	*p* < 0.001	*p* < 0.001	*p* < 0.001	*p* < 0.001			
Length (cm)	1	79.83 ± 2.55 (7.82) ^c^	85.86 ± 2.33 (7.19) ^b,c^	95.71 ± 0.75 (2.06) ^a^	89.14 ± 1.24 (3.69) ^a,b^	89.57 ± 1.7 (5.03) ^a,b^	89.00 ± 1.35 (4.00) ^a,b^	0.144	*p* < 0.001	*p* < 0.001
	2	93.83 ± 2.8 (7.30) ^b,c^	91.29 ± 2.20 (6.38) ^c^	102.14 ± 0.77 (1.99) ^a,b^	101.86 ± 3.09 (8.02) ^a,b^	103.86 ± 1.64 (4.18) ^a^	99.57 ± 0.81 (2.16) ^a,b,c^	*p* < 0.001	0.215	*p* < 0.01
	P	*p* < 0.001	*p* < 0.001	*p* < 0.001	*p* < 0.001	*p* < 0.001	*p* < 0.001			
Feed conversion ratio (%)	1	2.13 ± 0.15 (17.62) ^b^	2.35 ± 0.12 (13.30) ^a,b^	2.79 ± 0.13 (12.05) ^a^	2.35 ± 0.09 (9.88) ^a,b^	2.41 ± 0.1 (11.32) ^a,b^	2.41 ± 0.07 (7.42) ^a,b^	0.723	*p* < 0.01	*p* < 0.05
	2	3.05 ± 0.21 (16.73) ^b^	3.19 ± 0.17 (13.85) ^a,b^	3.58 ± 0.14 (10.31) ^a,b^	3.73 ± 0.13 (9.07) ^a^	3.51 ± 0.06 (4.20) ^a,b^	3.48 ± 0.5 (3.85) ^a,b^	*p* < 0.01	0.200	0.051
	P	*p* < 0.001	*p* < 0.001	*p* < 0.001	*p* < 0.001	*p* < 0.001	*p* < 0.001			

^1^ T1: conventional fed with no corn replacement; T2: corn replacement with 32% of cassava + taro; and T3: corn replacement with 42% of cassava + taro. ^a,b,c,d^ for each control least square means without a common superscript differ significantly (*p* < 0.05) between groups.

**Table 6 animals-13-00356-t006:** Chemical analysis of ileum content (mean ± standard error (coefficient of variation)) in backyard pigs fed with cooked cassava and taro in two locations in Ecuador (Quinindé and Río Chico).

Traits	Quinindé			Río Chico			P		
	Treatments ^1^								
	T1 (Control)	T2 (32%)	T3 (42%)	T1 (Control)	T2 (32%)	T3 (42%)	Location (L)	Treatment (T)	L × T
Humidity (%)	88.15 ± 1.42 (3.61)	84.15 ± 1.01 (5.05)	81.92 ± 1.91 (7.87)	84.86 ± 0.96 (2.54)	82.55 ± 0.81 (2.21)	83.97 ± 0.93 (2.47)	0.419	0.093	0.260
Protein (%)	6.08 ± 1.19 (19.51)	5.72 ± 0.25 (9.72)	6.72 ± 0.40 (13.24)	5.53 ± 0.42 (16.94)	6.88 ± 0.22 (7.11)	5.51 ± 0.43 (17.41)	0.536	0.455	*p* < 0.05
Fat (%)	0.98 ± 0.11 (24.11)	0.89 ± 0.06 (14.28)	1.13 ± 0.12 (23.88)	1.08 ± 0.06 (12.21)	1.27 ± 0.40 (70.68)	1.07 ± 0.45 (93.39)	0.504	0.961	0.700
Ash (%)	1.45 ± 0.19 (29.49)	2.08 ± 0.32 (34.53)	3.31 ± 0.70 (47.38)	1.98 ± 0.27 (30.18)	2.5 ± 0.77 (33.10)	2.68 ± 0.44 (36.45)	0.746	*p* < 0.05	0.318
Crude fiber (%)	2.69 ± 1.37 (113.59)	5.02 ± 1.83 (81.52)	6.21 ± 1.80 (64.8)	4.61 ± 1.02 (49.50)	4.07 ± 1.25 (68.51)	5.84 ± 0.40 (15.25)	0.859	0.234	0.549
Calories (kcal/100g)	35.77 ± 1.17 (7.32)	37.55 ± 2.08 (12.41)	39.88 ± 3.23 (18.09)	35.94 ± 1.06 (6.60)	37.66 ± 0.52 (3.10)	35.5 ± 4.85 (30.52)	0.529	0.734	0.614
Carbohydrates (%)	0.66 ± 0.35 (120.56)	1.57 ± 0.74 (105.18)	1.19 ± 0.45 (85.06)	1.08 ± 0.27 (56.08)	0.75 ± 0.19 (56.34)	0.93 ± 0.19 (45.92)	0.524	0.775	0.333
pH	6.09 ± 0.20 (7.30)	6.11 ± 0.21 (7.62)	5.92 ± 0.33 (12.64)	6.01 ± 0.26 (9.58)	6.81 ± 0.15 (4.94)	6.09 ± 0.27 (9.94)	0.192	0.143	0.279
Dry matter (%)	11.85 ± 1.42 (26.83)	15.37 ± 1.91 (27.84)	18.08 ± 2.88 (35.67)	15.14 ± 0.96 (14.22)	17.45 ± 0.81 (10.41)	16.03 ± 0.93 (12.92)	0.420	0.093	0.260

^1^ T1: conventional fed with no corn replacement; T2: corn replacement with 32% of cassava + taro; T3: corn replacement with 42% of cassava + taro. Ash is the powdery residue left after the burning of a substance.

**Table 7 animals-13-00356-t007:** Analysis of apparent ileum digestion (mean values are of (%) protein content) in backyard pigs from Ecuador fed with alternative cooked cassava and taro diets.

Traits ^1^	Quinindé			Río Chico			P		
	Treatments ^2^								
	T1 (Control)	T2 (32%)	T3 (42%)	T1 (Control)	T2 (32%)	T3 (42%)	Location (L)	Treatment (T)	L × T
NI–NH	9.22 ± 0.53	9.58 ± 0.25	8.58 ± 0.40	9.77 ± 0.42	9.79 ± 0.43	8.42 ± 0.22	0.536	*p* < 0.05	0.666
((NI–NH)/NI)	0.61 ± 0.04	0.64 ± 0.02	0.57 ± 0.03	0.65 ± 0.03	0.65 ± 0.03	0.56 ± 0.01			
Coefficient of digestibility	61.47 ± 3.54	63.87 ± 1.66	57.20 ± 2.65	65.13 ± 2.79	65.27 ± 2.86	56.13 ± 1.46			

^1^ NI = nutrient ingested and NH = nutrient in ileum ^2^ T1: conventional fed with no corn replacement; T2: corn replacement with 32% of cassava + taro; T3: corn replacement with 42% of cassava + taro.

**Table 8 animals-13-00356-t008:** Cost and benefit/cost (USD) of each pig group fed with alternative feeding based on cooked cassava and taro.

Activities	Quinindé			Río Chico		
	T1 (Control)	T2 (32%)	T3 (42%)	T1 (Control)	T2 (32%)	T3 (42%)
Purchase of pigs	350	350	350	420	420	420
Piggery (for rent)	100.00	100.00	100.00	26.44	26.44	26.44
Cost of feed	776.42	706.50	719.00	775.94	695.57	709.23
Vitamin AD3E	5.67	5.67	5.67	3.33	3.33	3.33
Vitamin B complex	2.20	2.20	2.20	2.33	2.33	2.33
Antidiarrheal (Diafin N Koning)	3.50	3.50	3.50	2.50	2.50	2.50
Probiovet^®^ (animal probiotic)	1.17	1.17	1.17	3.33	3.33	3.33
Ivermic simple 50 mL antiparasitic	1.67	1.67	1.67	1.67	1.67	1.67
Vaccione + earring	14.00	14.00	14.00	14.00	14.00	14.00
Syringes	0.42	0.42	0.42	1.75	1.75	1.75
Transport of materials	1.00	1.00	1.00	22.33	22.33	22.33
Other expenditure	120.00	120.00	120.00	60.00	60.00	60.00
Total cost	1376.03	1303.45	1299.45	1334.00	1253.00	1267.00
Kilo of meat produced	373.10	346.85	419.09	410.45	381.82	390.91
Gross income (pig sales)	2052.05	1907.67	2305.00	2,256	2099	2150
Net income	676.02	604.23	1005.55	923	846	883
Benefit/cost ratio	1.49	1.46	1.77	1.69	1.67	1.70
Unit costs per kilo of meat	3.88	3.91	3.61	3.68	3.69	3.67

T1: conventional fed with no corn replacement; T2: corn replacement with 32% of cassava + taro; T3: corn replacement with 42% of cassava + taro.

## Data Availability

This is not applicable, as the data are not in any data repository with public access. However, if an editorial committee needs access, we will happily provide them with it. Please use this email: erodero@uco.es.

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
