# Peer review of "Utilization of Cooked Cassava and Taro as Alternative Feed in Enhancing Pig Production in Ecuadorian Backyard System"

_animals, 2023, doi:10.3390/ani13030356_

Round 1

Reviewer 1 Report

The manuscript entitled "Productive results following employment of cooked cassava and taro as alternative feed in Ecuadorian pigs reared under 3 backyard system" has some novelty. The use of alternative feedstuff in animal production is a necessary in the world presently. 

There are some issues that must be addressed before the paper is published.  Some of these issues have been highlighted in the attached document and some comments have been placed below:

The introduction is too long and should be truncated to give a concise background as well as the objectives of the study. 

There are some major issues in the results and discussion section. The information should be provided to note differences in treatment based on growers and finishers as well as separate tables on the location effect. The digestibility should also be presented in this way. In this way readers can understand the effect of location and the effect of diet (T1-3). The column termed 'control of growth' is quite confusing.  The table on the economic information was well presented. 

Major revisions are needed for the presentation of data. P values should also be included. 

Author Response

RESPONSE

We would like to thank for your work and valuable comments that have substantially helped us to improve the former manuscript quality. All your comments have been considered and corrections have been made according to them. To make your work easier, we have highlighted the changes you and rest of reviewers requested by using track changes function. Also, the manuscript has been revised by a native speaker to improve the English.

We hope that you like the new version of the manuscript.

Thank you very much for your interest.

Kind Regards,

The authors

Reviewer 2 Report

Please see attached PDF FILE with comments on the manuscript. 

Author Response

(The authors gave the same response as above.)

Round 2

Reviewer 1 Report

The manuscript can be published based on changes made. However, the author should have provided a response to comments to upload to highlight where changes were made. 

Author Response

We would like to thank for your work and valuable comments that have substantially helped us to improve the former manuscript quality. All your comments have been considered and corrections have been made according to them. In this response letter we have highlight where changes were made. To make your work easier, we have highlighted the changes you and rest of reviewers requested by using track changes function. Also, the manuscript has been revised by a native speaker to improve the English.

We hope that you like the new version of the manuscript.

Thank you very much for your interest.

Kind Regards,

The authors

Reviewer 2 Report

A few more edits are attached in PDF File. Good job by the authors to address previous comments. 

- All Tables change Witness for Control

-Table 7 needs style revision

Line 381 - change food for feed

Author Response

We would like to thank for your work and valuable comments that have substantially helped us to improve the former manuscript quality. All your comments have been considered and corrections have been made according to them. To make your work easier, we have highlighted the changes you and rest of reviewers requested by using track changes function. Also, the manuscript has been revised by a native speaker to improve the English.

We hope that you like the new version of the manuscript.

Thank you very much for your interest.

Kind Regards,

The authors
